# Methylprednisolone Promotes *Mycobacterium smegmatis* Survival in Macrophages through NF-κB/DUSP1 Pathway

**DOI:** 10.3390/microorganisms11030768

**Published:** 2023-03-16

**Authors:** Anlong Li, Yonglin He, Chun Yang, Nan Lu, Jiajia Bao, Sijia Gao, Felycia Fernanda Hosyanto, Xintong He, Huichao Fu, Huajian Yan, Ningyu Ding, Lei Xu

**Affiliations:** 1Department of Pathogenic Biology, College of Basic Medicine, Chongqing Medical University, Chongqing 400016, China; 2Department of Clinical Medicine, Chongqing Medical University, Chongqing 400016, China; 3International Medical College, Chongqing Medical University, Chongqing 400016, China

**Keywords:** methylprednisolone, mycobacteria, NF-κB, DUSP1, ROS, IL-6

## Abstract

Background: *Mycobacterium tuberculosis* (*M. tuberculosis*) is the causative agent of tuberculosis. As an important component of host immunity, macrophages are not only the first line of defense against *M. tuberculosis* but also the parasitic site of *M. tuberculosis* in the host. Glucocorticoids can cause immunosuppression, which is considered to be one of the major risk factors for active tuberculosis, but the mechanism is unclear. Objective: To study the effect of methylprednisolone on the proliferation of mycobacteria in macrophages and try to find key molecules of this phenomenon. Methods: The macrophage line RAW264.7 infected by *M. smegmatis* was treated with methylprednisolone, and the intracellular bacterial CFU, Reactive Oxygen Species (ROS), cytokine secretion, autophagy, and apoptosis were measured. After the cells were treated with NF-κB inhibitor BAY 11-7082 and DUSP1 inhibitor BCI, respectively, the intracellular bacterial CFU, ROS, IL-6, and TNF-α secretion were detected. Results: After treatment with methylprednisolone, the CFU of intracellular bacteria increased, the level of ROS decreased, and the secretion of IL-6 and TNF-α decreased in infected macrophages. After BAY 11-7082 treatment, the CFU of *M. smegmatis* in macrophages increased, and the level of ROS production and the secretion of IL-6 by macrophages decreased. Transcriptome high-throughput sequencing and bioinformatics analysis suggested that DUSP1 was the key molecule in the above phenomenon. Western blot analysis confirmed that the expression level of DUSP1 was increased in the infected macrophages treated with methylprednisolone and BAY 11-7082, respectively. After BCI treatment, the level of ROS produced by infected macrophages increased, and the secretion of IL-6 increased. After the treatment of BCI combined with methylprednisolone or BAY 11-7082, the level of ROS produced and the secretion of IL-6 by macrophages were increased. Conclusion: methylprednisolone promotes the proliferation of mycobacteria in macrophages by suppressing cellular ROS production and IL-6 secretion through down-regulating NF-κB and up-regulating DUSP1 expression. BCI, an inhibitor of DUSP1, can reduce the level of DUSP1 in the infected macrophages and inhibit the proliferation of intracellular mycobacteria by promoting cellular ROS production and IL-6 secretion. Therefore, BCI may become a new molecule for host-directed therapy of tuberculosis, as well as a new strategy for the prevention of tuberculosis when treated with glucocorticoids.

## 1. Introduction

Tuberculosis is an infectious disease caused by *Mycobacterium tuberculosis* (*M. tuberculosis*). According to the latest Tuberculosis Report 2021 released by the World Health Organization, the most visible impact of the COVID-19 epidemic is a sharp drop in the number of newly diagnosed and reported tuberculosis cases globally, and the most immediate consequence is an increase in the number of people who will die from tuberculosis globally in 2020 [1]. By addressing the determinants of tuberculosis such as poverty, malnutrition, HIV infection, smoking, and diabetes, infections and severe illness can also be reduced, therefore reducing the number of mortalities from tuberculosis [1].

The natural history of tuberculosis begins with the inhalational transmission, hereafter, immunosuppressing the replication of *M. tuberculosis*. The result of this process is an asymptomatic latent tuberculosis infection (LTBI). According to the WHO, there are about 2–3 billion people in the world who are latently infected with *M. tuberculosis*; of these, 5–15% will suffer a relapse of TB during their lifetime [2]. It has also been reported that approximately 5–10% of normally immune individuals latently infected with *M. tuberculosis* will develop active tuberculosis during their lifetime [3]. In the United States of America, more than 80% of active tuberculosis cases are the result of LTBI activation, and most of these cases are preventable [4].

To eradicate tuberculosis, the WHO recommends screening and treating LTBI patients who are at risk of immunosuppression, for instance, by using immunosuppressive drugs, including glucocorticoids, disease-modifying antirheumatic drugs (immunosuppressants), TNF-α inhibitors, etc. [5]. Glucocorticoids are an extremely important class of regulatory molecules in the body, which is central to regulating the development, growth, metabolism and immune function of the body. It is the most important regulatory hormone of the body’s stress response, as well as the most widely effective anti-inflammatory and immunosuppressant used clinically. Oral or inhaled glucocorticoids have impaired the systemic or local immunity and are considered a major risk factor for active tuberculosis. Several reports have found an association between glucocorticoid use and the development of tuberculosis [6,7,8,9,10,11,12,13]. There are numerous etiologies of LTBI. The clinical use of drugs for high-risk groups is a major aspect. Through the investigation of related reports, it was found that glucocorticoids caused immune downregulation. Studying the specific mechanism of action can provide strategies for preventing the reactivation of LTBI and the treatment of tuberculosis.

In 1884, Lustgarten isolated a fast-growing mycobacterium (*Mycobacterium smegmatis*). *Mycobacterium smegmatis* (*M. smegmatis*) is widely distributed in soil, water, and plants. This study used *M. smegmatis* as a model for endoparasites in macrophages. Although the complete *M. smegmatis* genome sequence is still not available, available data show that 12 of the 19 pathogenic properties of *M. tuberculosis* are homologous to *M. smegmatis* and that *M. smegmatis* is a good model for studying the general biology of mycobacteria [14].

## 2. Materials and Methods

### 2.1. Culture of RAW 264.7 Cells

DMEM (high glucose) medium (Hyclone, Logan, UT, USA) with 10% FBS (Gbico, Waltham, MA, USA) was used for the culture of RAW 264.7 (ATCC TIB-71). The 6-well cell culture plates were inoculated with 1 × 10^6^ cells per well for bacterial infection.

### 2.2. Culture of M. smegmatis 

*M. smegmatis* MC^2^ 155 strain was inoculated in Middlebrook 7H9 Broth Base (BD, America) with a temperature of 37 °C and incubation time of 24–48 h in a shaking bed. Then, the cultured bacterial suspensions were used to infect RAW264.7 cells at MOI = 0.1.

### 2.3. Colony Forming Unit (CFU) Assay

RAW264.7 cells were infected by *M. smegmatis* (MOI = 0.1); after incubation at 37 °C for 1 h, the medium was discarded, the extracellular *M. smegmatis* was removed, DMEM (high glucose) medium with 10% FBS was then readded, and methylprednisolone (MCE, Dallas, TX, USA), BAY 11-7082 (MCE, USA), and BCI (MCE, USA) were added to incubate for 24- or 48-h-incubation. Subsequently, 1% Triton X-100 (Solarbio, Beijing, China) solution was added to cell culture plates, and the cells were lysed. Using the dilution coating method, the lysed suspension was inoculated on Middlebrook 7h10 agar medium (BD, Franklin Lakes, NJ, USA) supplemented with OADC, and colonies were counted after incubation at 37 °C in an incubator for 2–4 days.

### 2.4. Western Blot Analysis

RIPA Lysis Buffer (Beyotime, Shanghai, China) containing 1% PMSF (Beyotime, China) and 1% phosphatase inhibitors (CWBIO, Taizhou, China) were added to the culture plates, cells were lysed on ice and total protein was collected. The protein solutions were separated by SDS polyacrylamide gel electrophoresis and transferred to PVDF membranes. PVDF membranes (Merck, Germany) were blocked using 5% skimmed milk for 2 h, primary antibody (HUABIO, Hangzhou, China) was incubated at 4 °C for more than 12 h, secondary antibody (HUABIO, China) was incubated for 2 h, and, subsequently, ECL chemiluminescence Kit (Beyotime, China) was used for the development.

### 2.5. Flow Cytometry

Cells in the cell culture plate were collected into centrifuge tubes, centrifuged at 1000× *g* r/min for 5 min, and had the supernatant discarded and resuspended in PBS (Hyclone, USA) twice. Then, all cells were resuspended in 500 μL PBS (pH = 7.2) buffer and placed in a 1.5 mL EP tube for subsequent experiments. Annexin V-FITC/PI Apoptosis Detection Kit (MCE, USA) was used for apoptosis detection, and the reactive oxygen species assay kit (Beyotime, China) was used for ROS detection.

### 2.6. Transcriptome High-Throughput Sequencing

Total cellular RNA was extracted by Trizol ^®^ Reagent (Invitrogen, Waltham, MA, USA), and the concentration and purity of the extracted RNA were checked by nanodrop2000 (Thermo Fisher Scientific, Waltham, MA, USA). A single library building requires a total amount of RNA greater than 1 μg, concentrations ≥ 50 ng/μL, and OD 260/280 between 1.8 and 2.2. mRNA was isolated from total RNA and used to analyze the transcriptome information. Eukaryotic mRNA sequencing was based on the hiseq platform, and sequencing experiments were performed using the Illumina Truseq™ RNA sample prep Kit for library construction.

### 2.7. Screening of Differentially Expressed Genes (DEGs)

DEGs screening was performed using the software DEseq2, which is based on a negative binomial distribution model by calculating the differential expression from read count data aligned to the gene. The filtering criteria for significantly DEGs were FDR < 0.05 & |log2FC|> 1, and a gene was considered as differentially expressed when both conditions were met.

### 2.8. Construction of a Protein–Protein Interaction (PPI) Network

PPI network analysis was based on the protein information corresponding to the gene. PPI analysis was performed using the STRING database (http://string-db.org/, accessed on 7 July 2022). After obtaining the PPI network data, networkX under Python was used to visualize the network of interested genes. By analyzing the topological properties of the PPI network, such as the degree, betweenness, closeness, and cluster coefficient of each node in the network, the key nodes in the interaction network can be obtained.

### 2.9. Analysis of the Kyoto Encyclopedia of Genes and Genomes (KEGG)

The analysis used KOBAS (http://kobas.cbi.pku.edu.cn/home.do/, accessed on 7 July 2022) for KEGG pathway enrichment analysis and Fisher’s exact test for calculation. To control the calculation of the false positive rate, the BH(FDR) method was used for multiple testing. The corrected *p* value was set as the threshold of 0.05, and the KEGG pathways that met this condition were defined as those significantly enriched in the DEGs.

### 2.10. Analysis of Gene Ontology (GO)

The software Goatools was used for GO enrichment analysis using Fisher’s exact test. To control the calculated false positive rate, *p* values were adjusted using four multiple testing methods (Bonferroni, Holm, BH, and BY). Generally, the GO function was considered to be significantly enriched when the FDR ≤ 0.05.

### 2.11. Construction of the Heat Map

Transcripts with similar expression patterns are often functionally relevant, and this module performs expression pattern clustering analysis of transcripts in the selected gene set. Based on the expression information of transcripts in different samples, the distance between transcripts was calculated, and the transcripts were then classified by the iterative method. Statistical analysis and visualization were performed in R version 3.6.3, involving the R package FASTCLUSTER.

### 2.12. Eenzyme Linked Immunosorbent Assay (ELISA)

The cell culture supernatant was collected, and TNF-α and IL-6 levels were measured. ELISA was performed using a commercial kit, Mouse TNF-alpha ELISA Kit (ABclonal, Wuhan, China), and Mouse IL-6 ELISA Kit (ABclonal, China).

### 2.13. Quantitative Reverse Transcription Polymerase Chain Reaction (qRT-PCR)

Total RNA was extracted using TRIzol^®^ Reagent (Invitrogen, USA), and the concentration and purity of the extracted RNA were examined using a spectrophotometer (IMPLEN, München, Germany). mRNAs were reverse transcribed into cDNA templates using the Reverse Transcription Kit (Takara, Gunma, Japan), and the reverse transcription reaction conditions were 37 °C for 15 min, 85°C for 5 s, and ended at 4°C. In the next step, cDNA quantification was performed using SYBR^®^ Premix Ex Taq™ Kit (Takara, Japan) and CFX96™ real time PCR system (BIO RAD, Hercules, CA, USA). The program was set as the first step of predenaturation at 95 °C for 30 s, the second step was 40 cycles of 95 °C for 5 s, and 60 °C for 30 s. The qRT-PCR primer sequences are shown in Table 1.

### 2.14. Data Processing 

Graphpad prism 8 was used for data processing. The t-test was used to analyze the data of the two groups. The analysis of variance was used for data analysis of more than two groups. Ns means no statistical difference, “*” means *p* value < 0.05, “**” means *p* value < 0.01, “***” means *p* value < 0.001, and “****” means *p* value < 0.0001.

### 2.15. Assay of Cell Viability 

CCK-8 assay was used to measure cell viability; CCK-8 commercial kit (MCE, USA) was used for the experiments. OD was determined at 450 nm wavelength.

## 3. Results

### 3.1. Methylprednisolone Facilitates M. smegmatis Survival in Macrophages and Suppresses Cellular Secretion of TNF-α and IL-6

The infected cells were treated with methylprednisolone and cultured for 48 h. The cell supernatant was collected for the quantitative detection of inflammatory factors, which showed that methylprednisolone could inhibit the secretion of TNF-α and IL-6 from infected cells (Figure 1A,B). The infected cells were treated with methylprednisolone, and the number of colonies was counted after culture of 24 and 48 h. It was found that the number of colonies increased with the increase in methylprednisolone concentration (Figure 1C,D).

### 3.2. Methylprednisolone Promotes M. smegmatis Survival in Macrophages by Suppressing the Expression of NF-κB

The infected cells were treated with methylprednisolone and BAY 11-7082 (an NF-κB inhibitor) for 24 and 48 h. Western blot analysis showed that methylprednisolone and BAY 11-7082 inhibited the expression of NF-κB and promoted the expression of IκB-α (Inhibitor of NF-κB) (Figure 2A). After confirming that methylprednisolone inhibited the expression of NF-κB, the cells infected with *M. smegmatis* were treated with BAY 11-7082 for 24 and 48 h, and the intracellular *M. smegmatis* colonies count was performed to confirm that methylprednisolone promoted the survival of *M. smegmatis* in macrophages by inhibiting NF-κB (Figure 2B).

### 3.3. Methylprednisolone Promotes M. smegmatis Survival in Macrophages, but Not by Inhibiting Infected Cell Apoptosis

Infected cells were treated with methylprednisolone. After 48 h of culture, cells were collected, and the apoptotic cell ratio was detected by flow cytometry. The results showed that the apoptotic rate of *M. smegmatis*-infected cells did not change significantly after methylprednisolone treatment (Figure 3A). The infected cells were collected and then subjected to Western blot analysis, which showed that there was no difference in the apoptotic-related protein p53 and its phosphorylated protein (Figure 3B).

### 3.4. Methylprednisolone Suppresses Cellular ROS Production through Inhibiting the Expression of NF-κB

Flow cytometry analysis of ROS showed that cellular ROS production in the infected cells was decreased after treatment with methylprednisolone (Figure 4A) and BAY 11-7082 (Figure 4B). These results suggest that methylprednisolone inhibits ROS production in infected cells by inhibiting the expression of NF-κB. 

### 3.5. High-Throughput Sequencing Analyzed the Expression of Genes in M. smegmatis Infected Macrophages after BAY 11-7082 Treatment

*M. smegmatis* infected macrophages were treated with BAY 11-7082 for 24 h and then subjected to transcriptome high-throughput sequencing analysis. Based on the quantitative results of expression, the DEGs between the two groups were obtained. The screening threshold was as follows: |log2FC| > 1.00 & P. adj < 0.05, a total of 164 DEGs were identified, including 54 up-regulated genes and 110 down-regulated genes (Figure 5A), and all DEGs are detailed in Appendix A. According to the comprehensive score greater than 0.4, the top 100 DEGs were screened, and the discrete gene nodes were deleted to obtain the PPI network diagram (Figure 5B), which showed that TNF was the hub gene.

The DEGs were clustered into 10 categories (Figure 5C), and the first two subsets with the largest number of gene clusters (150 genes, 90 genes) were drawn to plot the subclustering trend (Figure 5D,E), and the clustering results showed that the differences within the group were relatively stable.

Based on the KEGG database, gene function enrichment analysis was performed, and Figure 5F was obtained. The results showed that cytokine–cytokine receptor interaction (9 genes), MAPK signaling pathway (9 genes), and Kaposi sarcoma-associated herpesvirus infection (7 genes) were the top three pathways with gene enrichment. Based on the GO database, gene function enrichment analysis was performed and Figure 5G was obtained. The results showed that biological regulation (162 genes), regulation of biological process (153 genes), and regulation of cellular process (144 genes) were the top three gene enrichment pathways.

### 3.6. Methylprednisolone and BAY 11-7082 Up-Regulate the Expression of DUSP1

Infected cells were treated for 48 h, the DEG DUSP1 (Figure 5A,B) was verified by qRT-PCR (Figure 6A), and Western blot analysis also confirmed the expression level of DUSP1 (Figure 6B). We found that methylprednisolone and BAY 11-7082 promoted the expression of DUSP1 in macrophages infected with *M. smegmatis*.

### 3.7. Low Level of DUSP1 Inhibited the Survival of M. smegmatis in Macrophages through NF-κB/DUSP1 Pathway

BCI is an inhibitor of DUSP1. BCI was used to inhibit DUSP1 in infected cells (Figure 7A,B), and then the number of *M. smegmatis* in macrophages was observed. After diluting the concentration gradient of BCI, RAW264.7 cells were treated for 48 h. CCK-8 assay showed that 0.5–1 μM had little damage to cells (Figure 7C), and the cells were treated with a concentration at most 1 μM in the following experiments. The macrophages infected with *M. smegmatis* were treated with BCI at a concentration of 0–1 μM for 24 and 48 h, and then the CFU was measured. The results showed that the CFU decreased gradually with the increase in BCI concentration (Figure 7D). After methylprednisolone treatment for 24 and 48 h, the CFU of *M. smegmatis* increased, and BCI could inhibit this phenomenon (Figure 7E). After BAY 11-7082 treatment for 24 and 48 h, the CFU of *M. smegmatis* increased, and BCI could inhibit this phenomenon (Figure 7F).

### 3.8. Inhibiting DUSP1 Prompts Cellular ROS Production and IL-6 Secretion in Macrophages Infected with M. smegmatis 

BCI could promote ROS production in infected macrophages (Figure 8A). ROS production was inhibited in infected macrophages after methylprednisolone treatment, and BCI could inhibit the effect of methylprednisolone treatment (Figure 4A and Figure 8B). ROS production was inhibited in infected macrophages after BAY 11-7082 treatment, and BCI could inhibit the effect of BAY 11-7082 treatment (Figure 4B and Figure 8C).

BCI can promote the secretion of IL-6 in macrophages infected with *M. smegmatis*, methylprednisolone and BAY 11-7082 could inhibit the secretion of IL-6 in the infected macrophages, and BCI could inhibit the effect of methylprednisolone/BAY 11-7082 treatment (Figure 9A). However, the secretion of TNF-α was not the same as that of IL-6 (Figure 9B).

## 4. Discussion

When mycobacteria invade host cells, the cellular inflammatory response is activated. The host utilizes anti-inflammatory and pro-inflammatory interregulation to control infection. During the latent infection of *M. tuberculosis*, anti-inflammatory and pro-inflammatory are in balance. However, during the active phase of the disease, this homeostasis is disrupted, and disease progression occurs [15]. Pro-inflammatory responses and lung tissue remodeling against tuberculosis infection are important for bacterial eradication, but they may lead to excessive inflammation and persistent lung injury, and the remodeling of lung tissue or auxiliary regulation of inflammation may improve the outcome of tuberculosis treatment; anti-inflammatory responses, on the other hand, prevent tissue damage but may result in suboptimal bacterial clearance [15], and these responses are often mediated by immunosuppressive cell populations. Hence, it can improve the treatment effect of tuberculosis by regulating the pro-inflammatory response.

NF-κB is an essential transcription factor for the inflammatory response, and its activity is strictly regulated by a variety of mechanisms. It is involved in apoptosis, viral replication, tumorigenesis, inflammation, and numerous autoimmune diseases [16]. Activation of NF-κB is considered as part of the stress response because it can be activated by a variety of stimuli, including bacterial and viral infections, pro-inflammatory cytokines, antigen receptors, and so forth [17]. Macrophages are a large class of innate immune cells existing in different tissues and playing a role in the first line of fighting infection. The pro-inflammatory function of NF-κB has been extensively studied in macrophages. Innate immune cells express pattern recognition receptors (PRRs); for instance, toll-like receptors (TLRs) can identify various pathogenic microbial components, so-called pathogen-associated molecular patterns (PAMPs) [18]. A common signaling event of PRRs is the activation of the canonical NF-κB pathway, which is responsible for the induction of transcription of pro-inflammatory cytokines, chemokines, and other inflammatory mediators in different types of innate immune cells [18,19]. miRNA let-7 can regulate the immune response of *M. tuberculosis* infection by regulating the inhibitor A20 of the NF-kB pathway, suggesting that the regulation of A20 by miRNA let7f regulates the activity of NF-κB during *M. tuberculosis* infection. After the overexpression of A20, miRNA-let-7f-mediated NF-κB activity is increased and weakened during the infection with *M. tuberculosis*. This suggests that the miRNA let-7-A20 pathway directly regulates NF-κ B activity during infection [20]. In RAW264.7 and THP-1 cells, the *M. tuberculosis*-infected host significantly upregulates miR-125a in a TLR4-signal-dependent manner, which is a negative regulator of the NF-kB pathway. It directly targets TRAF6, thereby inhibiting cytokines, attenuating the immune response, and promoting the survival of *M. tuberculosis* [21]. Various studies have shown that the activity of NF-κB is closely related to the survival of mycobacteria and it is a potential therapeutic target for mycobacterial infections. Improper clinical use of glucocorticoids (such as methylprednisolone) can lead to LTBI. Our study found that inhibiting NF-κB can inhibit the body’s inflammatory response and disrupt the balance of anti-inflammatory and pro-inflammatory responses, thereby promoting the survival of mycobacteria in macrophages. 

DUSP1 is a member of the family of dual specificity phosphatases (DUSPs), which is also known as mitogen-activated protein kinase (MAPK) phosphatase 1 (MKP-1). It is a protein with anti-inflammatory properties. In the context of infectious diseases and sepsis, studies on the role of DUSP1 have focused on macrophage responses [22]. Although activation of TLRs is essential to elicit an acute inflammatory response and effectively clear pathogens, an excessive inflammatory response is dangerous to the host. To prevent harmful inflammation, a number of signaling mechanisms are induced, including downregulation of the surface abundance of TLRs and transcriptional induction of genes encoding negative regulatory proteins [23,24]. Among these feedback mechanisms, the rapid induction of DUSP1 expression inhibits the overactivation of MAPKs [25]. Studies of psoriasis or colitis have shown that down-regulation of DUSP1 promotes the occurrence of these diseases, and DUSP1 may be a target for disease treatment [26,27,28,29]. DUSP1-/- macrophages produce excess cytokines, e.g., interleukin-6 (IL-6) and IL-10 [30,31,32,33]. Dexamethasone inhibits the death of *M. tuberculosis*-infected cells by promoting DUSP1-dependent dephosphorylation of p38 MAPK; at the same time, dexamethasone combined with DUSP1-specific inhibitor BCI can completely block the protective effect of dexamethasone on *M. tuberculosis*-infected human lung fibroblasts [34]. Furthermore, our study found that BCI can inhibit DUSP1 induced by methylprednisolone and BAY 11-7082 in infected cells, and can reduce the number of mycobacteria in macrophages, which may be associated with the increased production of ROS and the release of IL-6.

Reactive oxygen species (ROS) contain partially reduced oxygen-containing molecules. They are free radicals and/or oxygen derivatives, including superoxide anion, hydrogen peroxide, hydroxyl radicals, lipid hydroperoxides, and peroxyl radicals [35]. ROS levels are in a state of dynamic equilibrium in the cell. Most intracellular ROS originate from superoxide radicals formed mainly through NADPH oxidases (NOXs), xanthine oxidases (XO), and mitochondrial electron transport chain (mETC) in endogenous biological systems [36,37]. Recognition of bacteria by macrophages leads to the generation of ROS in different cellular substructures where they fulfill different antimicrobial functions [38]. Recognition of invading bacteria induces rapid and robust production of ROS (extracellular ROS) in the extracellular space and lumen of the phagosome [39,40,41]. *M. tuberculosis* utilizes the macrophage phagosome as the site of proliferation. The survival mechanism in the phagosome is to inhibit the fusion of phagosomes with lysosomes and avoid being degraded and eliminated by autophagocytosis [42]. Our study found that the use of methylprednisolone and BAY 11-7082 can reduce the level of ROS produced by RAW264.7, which are infected by *M. smegmatis*, and BCI can inhibit the effect of methylprednisolone and BAY 11-7082. Eventually, methylprednisolone reduces ROS levels and promotes the survival of *M. smegmatis* in macrophages through the NF-κB/DUSP1 pathway. We also found that BCI treatment of *M. smegmatis*-infected macrophages reduces *M. smegmatis* survival. We speculate that BCI inhibits the level of DUSP1 and promotes ROS generation in macrophages, leading to the reduction of *M. smegmatis*. 

IL-6 is a key cytokine in infection, cancer, and inflammation. It drives disease progression or supports the maintenance of the immune response. Almost all stromal and immune system cells produce IL-6, whereas IL-1β and tumor necrosis factor are the main activators of IL-6 expression. Other pathways such as toll-like receptors, prostaglandins, adipokines, stress response, and other cytokines can also promote IL-6 synthesis [43]. The normal physiological concentration of IL-6 in human serum is relatively low (1–5 pg/mL). In contrast, in disease settings and extreme situations (e.g., meningococcal septic shock), IL-6 levels increase rapidly and can reach the mg/mL range [44]. *M. tuberculosis* controls the production of IL-6 in the host and inhibits type I interferon signaling, thus inhibiting disease progression, and suggesting that IL-6 secreted by *M. tuberculosis* infected macrophages may be involved in the cellular immune response against *M. tuberculosis* infection [45]. Tocilizumab is a novel monoclonal antibody that competitively inhibits the binding of interleukin-6 (IL-6) to its receptor (IL-6R) and inhibits the entire receptor complex to prevent IL-6 signaling and reduces inflammation levels [46]. Tocilizumab (TCZ) is used to treat severe COVID-19 caused by SARS-CoV-2 infection. Unintended consequences of TCZ treatment include reactivation of tuberculosis (TB) or hepatitis B virus (HBV) and hepatitis C virus (HCV) deterioration [47]. These evidence suggest that IL-6 levels are related to the activation of LTBI. Our study found that the use of methylprednisolone and BAY 11-7082 was able to reduce the level of IL-6 produced by *M. smegmatis* infected RAW264.7, and BCI can inhibit the effect of methylprednisolone and BAY 11-7082 and increase the level of IL-6. An interesting finding is that BCI can promote the production of ROS and the secretion of IL-6 without promoting the secretion of TNF-α (this can reduce host cell damage). Finally, we speculate that methylprednisolone reduces the level of IL-6 secretion and the inflammatory level of RAW264.7 through the NF-κB/DUSP1 pathway, hence promoting the survival of *M. smegmatis* in macrophages.

## 5. Conclusions

Methylprednisolone promotes the proliferation of mycobacteria in macrophages by suppressing cellular ROS production and IL-6 secretion through down-regulating NF-κB and up-regulating DUSP1 expression. BCI, an inhibitor of DUSP1, can reduce the level of DUSP1 in the infected macrophages and inhibit the proliferation of intracellular mycobacteria by promoting cellular ROS production and IL-6 secretion. At the same time, BCI does not increase TNF-α release and thus does not cause cellular damage. Therefore, BCI may become a new molecule for host-directed therapy of tuberculosis, as well as a new strategy for the prevention of tuberculosis when treated with glucocorticoids.

## Figures and Tables

**Figure 1 microorganisms-11-00768-f001:**
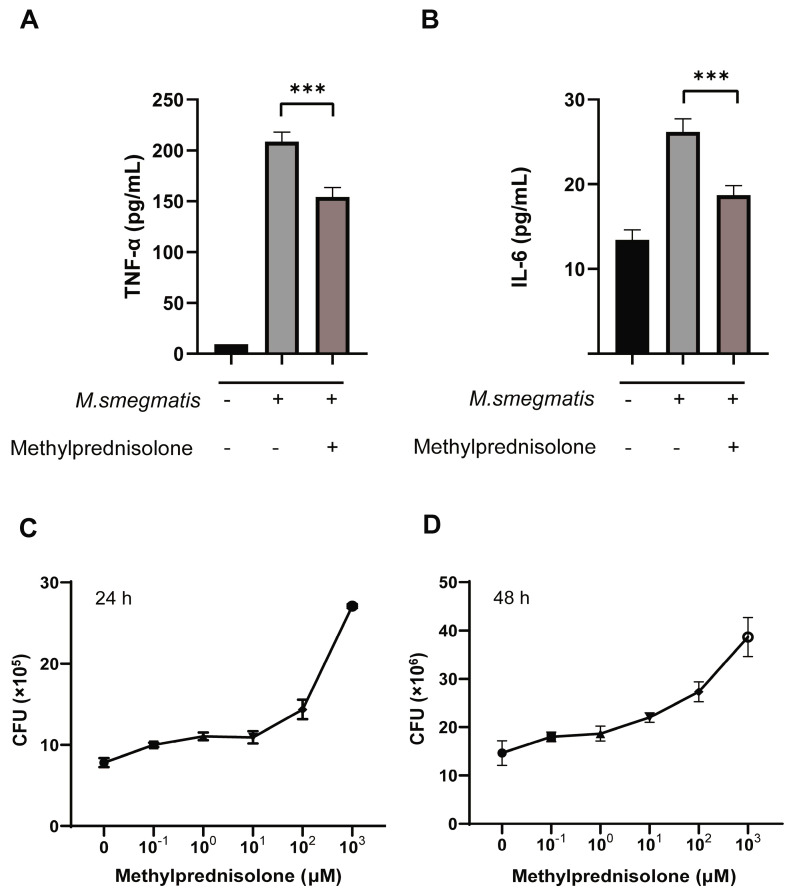
(**A**,**B**) Methylprednisolone (10^2^ μM) treated *M. smegmatis* infected RAW264.7 cells for 48 h; the secretion of TNF-α and IL-6; the t-test was used for data analysis; *** *p* < 0.001. (**C**,**D**) Infected RAW264.7 cells were treated with different concentrations of methylprednisolone for 24 and 48 h, and CFU were measured.

**Figure 2 microorganisms-11-00768-f002:**
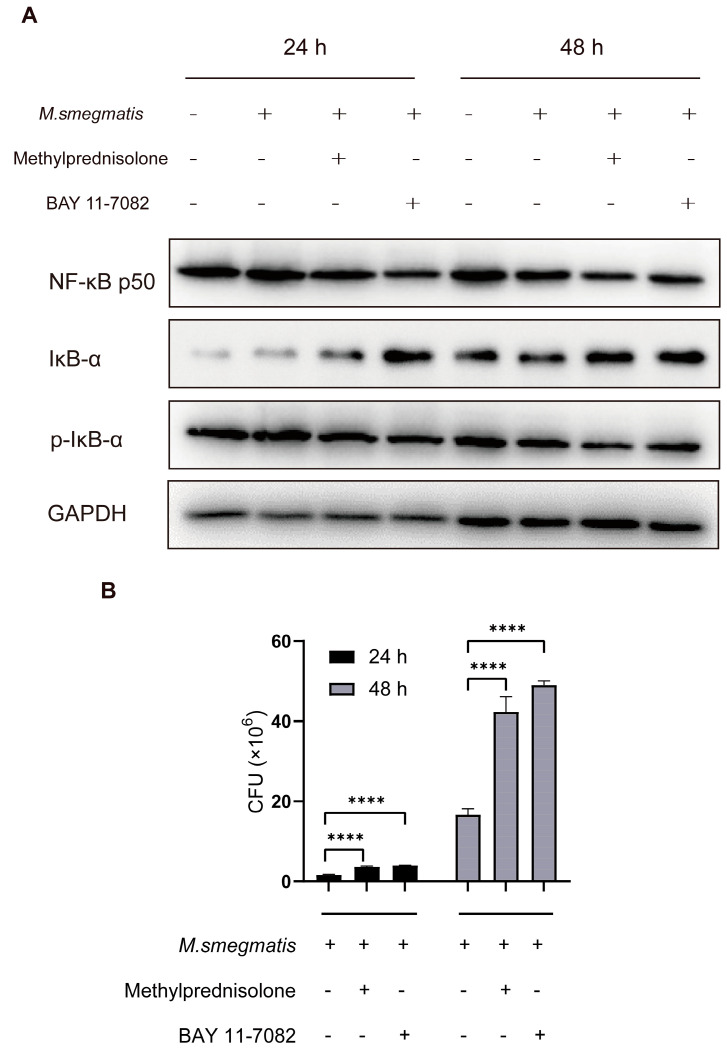
(**A**) Methylprednisolone (10^2^ μM) and the NF-κB inhibitor BAY 11-7082 (2 μM) treated *M. smegmatis* infected RAW264.7 cells for 24 and 48 h; NF-κB p50, IκB-α, and p-IκB-α levels were detected with Western blot analysis. (**B**) CFU results of RAW264.7 cells infected with *M. smegmatis* after treatment with methylprednisolone (10^2^ μM) and NF-κB inhibitor BAY 11-7082 (2 μM) for 24 h and 48 h; the analysis of variance was used for data analysis; **** *p* < 0.0001.

**Figure 3 microorganisms-11-00768-f003:**
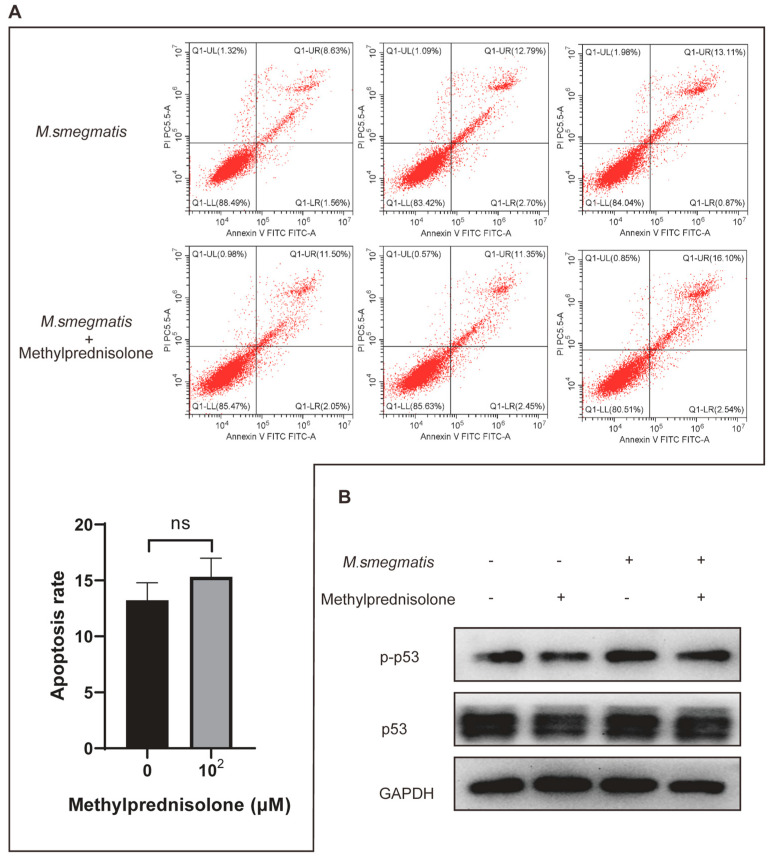
Methylprednisolone (10^2^ μM) treated *M. smegmatis* infected RAW264.7 cells for 48 h; (**A**) apoptotic cells were detected by flow cytometry; the t-test was used for data analysis; ^ns^ no statistical differences. (**B**) Apoptosis-related protein p53 and p-p53 level were detected with Western blot analysis.

**Figure 4 microorganisms-11-00768-f004:**
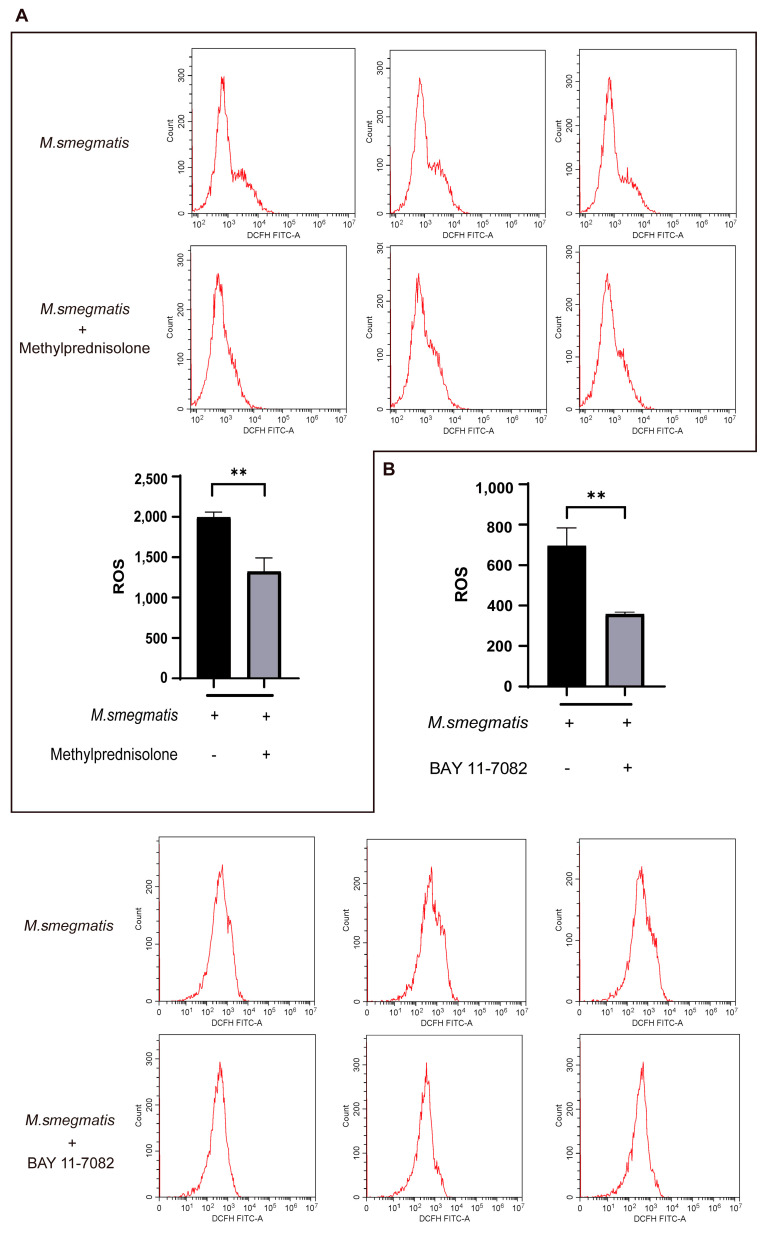
(**A**) Methylprednisolone (10^2^ μM) treated *M. smegmatis* infected RAW264.7 cells for 48 h; flow cytometry was used to detect the level of cellular ROS production; the t-test was used for data analysis; ** *p* < 0.01. (**B**) BAY 11-7082 (2 μM) treated *M. smegmatis* infected RAW264.7 cells for 48 h; flow cytometry was used to detect the level of cellular ROS production; the t-test was used for data analysis; ** *p* < 0.01.

**Figure 5 microorganisms-11-00768-f005:**
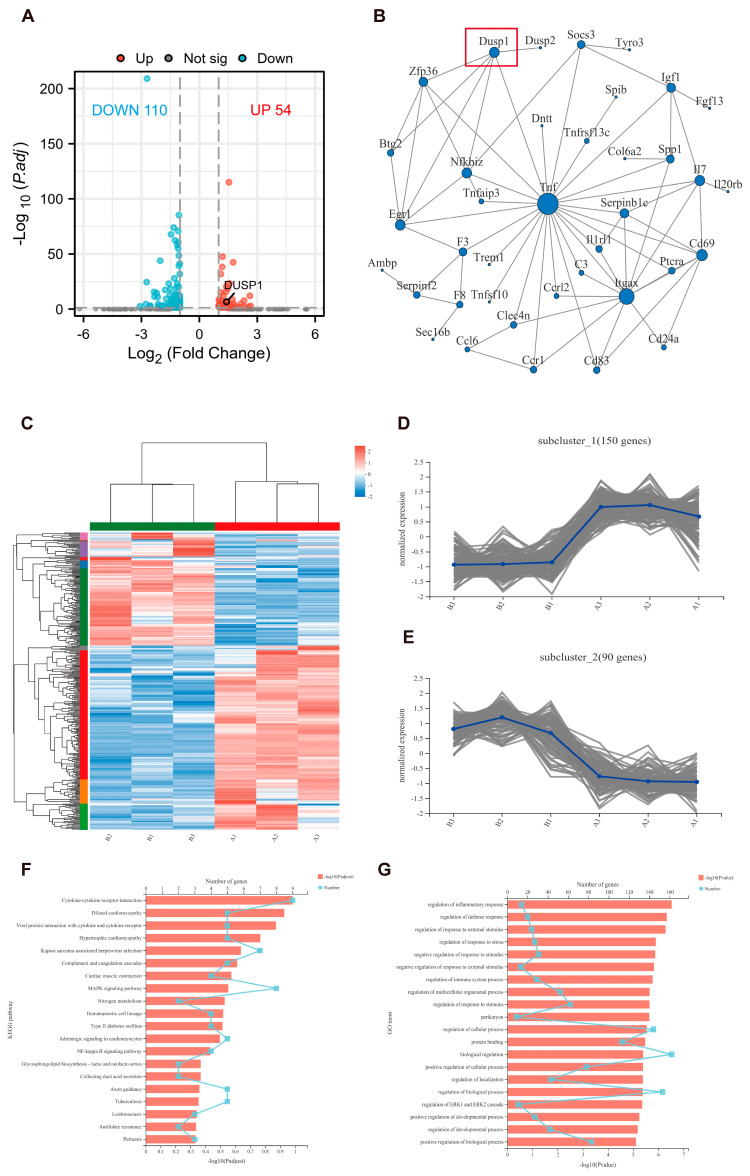
*M. smegmatis* infected RAW264.7 cells treated with BAY 11-7082 for 24 h. Transcriptome high-throughput sequencing analysis was performed. (**A**) Volcano diagram: based on the quantitative results of expression, the DEGs between the two groups were obtained. The blue was the lowly expressed genes in the treatment group, and the red was the highly expressed genes in the treatment group. (**B**) The PPI network: according to the comprehensive score greater than 0.4, the top 100 differentially expressed genes were screened, and discrete gene nodes were deleted to obtain PPI network. (**C**) Heat map: each column represents a sample (B1, B2, and B3 are BAY 11-7082 treated groups, and A1, A2, and A3 are control groups), each row represents a gene, and the color in the figure indicates the expression level of the gene/transcript in the sample; red represents the higher expression level of the gene/transcript in the sample, and the blue represents the lower expression level. The left side shows the gene/transcript clustering dendrogram, where the closer the two gene/transcript clades are, the closer their expression levels are. (**D**,**E**) Subcluster trend graph: he abscissa is the samples name (B1, B2, and B3 are the BAY 11-7082 treatment group, and A1, A2, and A3 are the control group), and the ordinate is the gene expression level in the samples, each line in the graph represents a gene change trend, and the blue line represents the change trend of the average expression of all gene transcripts. (**F**) Based on KEGG database, gene function enrichment analysis was performed; the lower abscissa represents log10 (*p* value), and the upper abscissa represents the number of enriched genes. (**G**) Based on GO database, gene function enrichment analysis was performed; the lower abscissa represents log10 (*p* value), and the upper abscissa represents the number of enriched genes.

**Figure 6 microorganisms-11-00768-f006:**
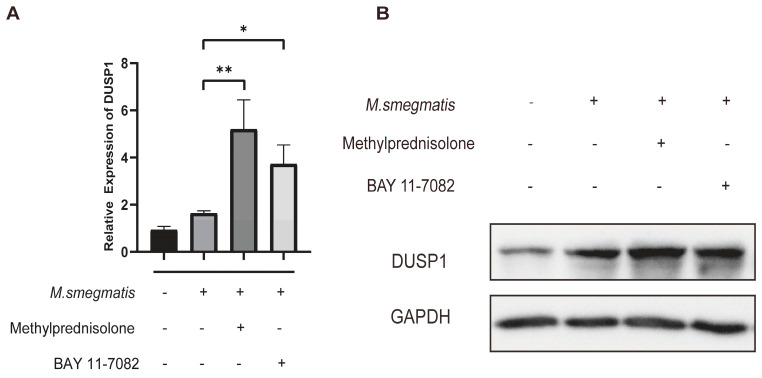
Methylprednisolone (10^2^ μM) treated *M. smegmatis* infected RAW264.7 cells for 48 h, cellular Total-RNA was verified by qRT-PCR, and total protein was extracted for Western blot analysis. (**A**) Relative expression of DUSP1 was detected with qRT-PCR, and the analysis of variance was used for data analysis; * *p* < 0.05; ** *p* < 0.01. (**B**) DUSP1 levels were detected with Western blot analysis.

**Figure 7 microorganisms-11-00768-f007:**
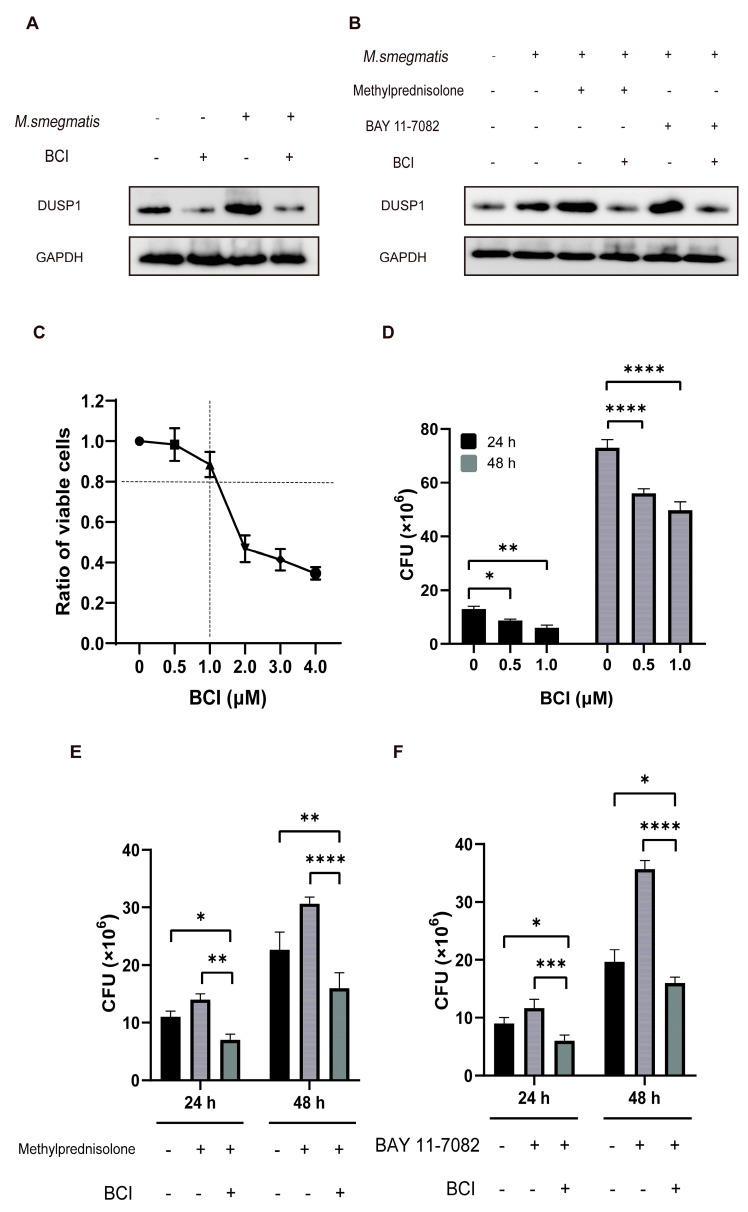
(**A**) After treatment of 48 h, DUSP1 levels were detected with Western blot analysis, cellular DUSP1 levels could be decreased by BCI (1 μM) treatment, (**B**) and BCI (1 μM) could reduce the level of DUSP1 elevated by methylprednisolone and BAY11-7082. (**C**) RAW264.7 cells were treated with different concentrations of BCI (0, 0.5, 1, 2, 3, 4 μM) for 48 h, and the cell viability was determined by the CCK-8 method. (**D**) RAW264.7 cells infected with *M. smegmatis* were treated with different concentrations of BCI (0, 0.5, 1 μM) for 24 and 48 h, and the number of intracellular *M. smegmatis* was measured; the analysis of variance was used for data analysis; * *p* < 0.05; ** *p* < 0.01; **** *p* < 0.0001. (**E**) RAW264.7 cells infected with *M. smegmatis* were treated with methylprednisolone (10^2^ μM) and BCI (1 μM) for 24 and 48 h, and the number of intracellular *M. smegmatis* was measured; the analysis of variance was used for data analysis; * *p* < 0.05; ** *p* < 0.01; **** *p* < 0.0001. (**F**) RAW264.7 cells infected with *M. smegmatis* were treated with BAY 11-7082 (2 μM) and BCI (1 μM) for 24 and 48 h, and the number of intracellular *M. smegmatis* was measured; the analysis of variance was used for data analysis; * *p* < 0.05; *** *p* < 0.001; **** *p* < 0.0001.

**Figure 8 microorganisms-11-00768-f008:**
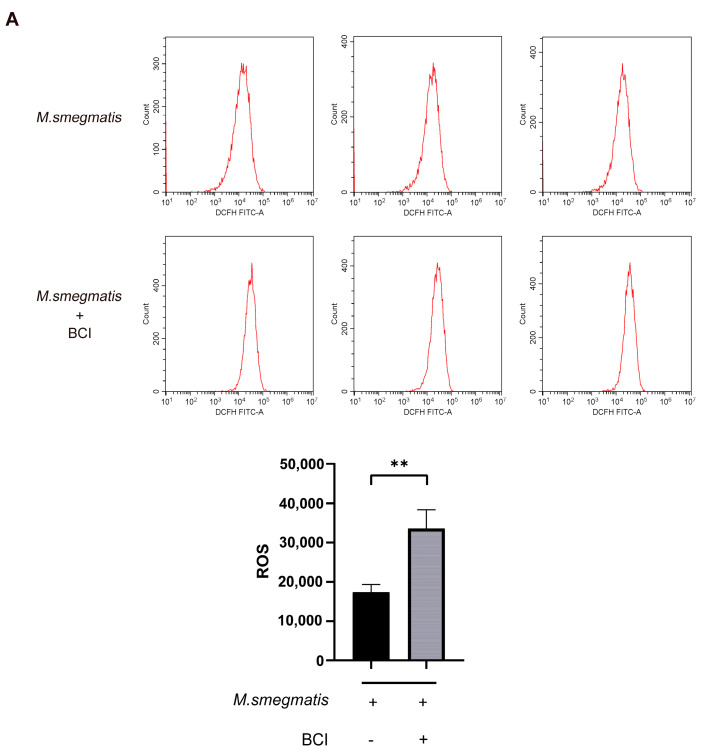
(**A**) BCI (1 μM) treated *M. smegmatis* infected RAW264.7 cells for 48 h; flow cytometry was used to detect the level of cellular ROS production; the t-test was used for data analysis; ** *p* < 0.01. (**B**) Methylprednisolone (10^2^ μM) and BCI (1 μM) treated *M. smegmatis* infected RAW264.7 cells for 48 h; flow cytometry was used to detect the level of cellular ROS production; the t-test was used for data analysis; ** *p* < 0.01. (**C**) BAY 11-7082 (2 μM) and BCI (1 μM) treated *M. smegmatis* infected RAW264.7 cells for 48 h; flow cytometry was used to detect the level of cellular ROS production; the *t*-test was used for data analysis; *** *p* < 0.001.

**Figure 9 microorganisms-11-00768-f009:**
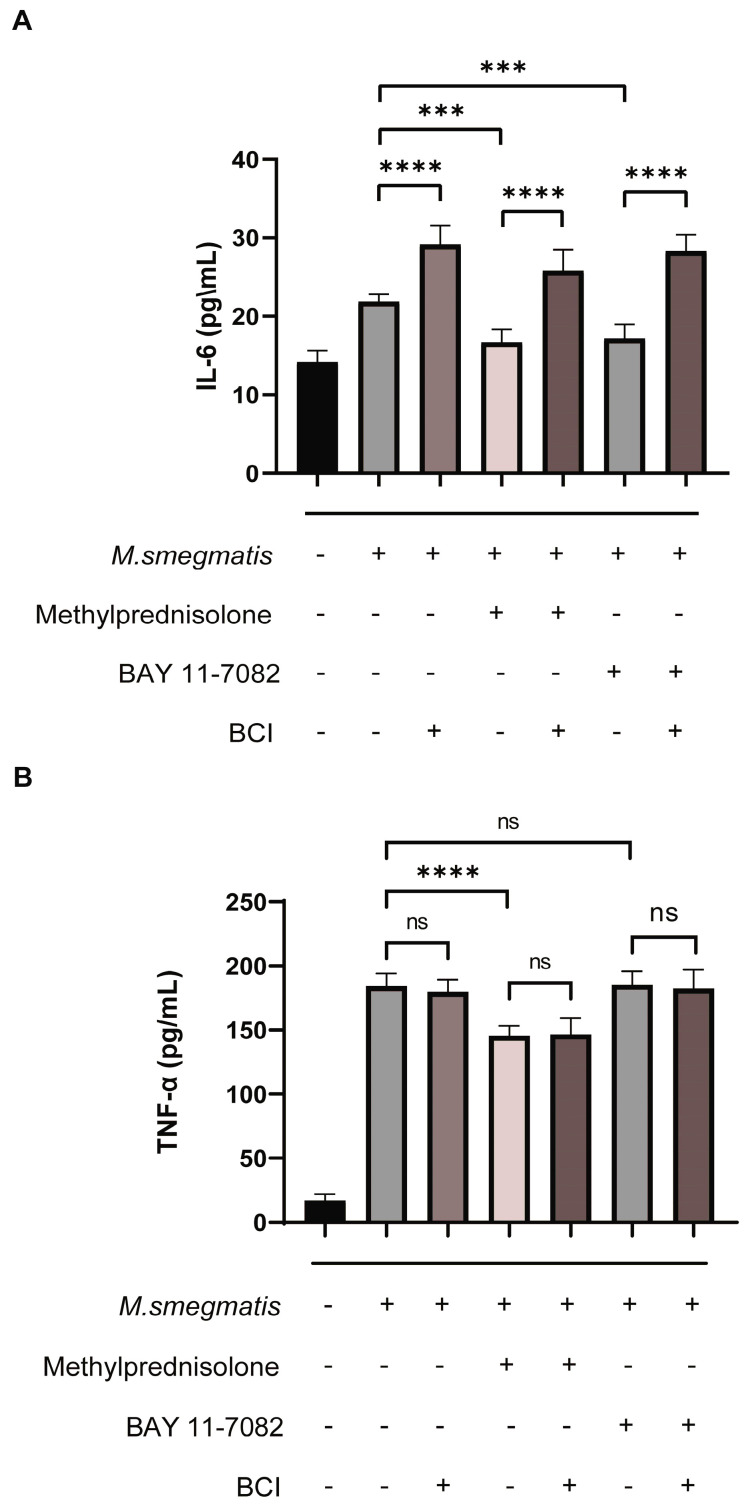
RAW264.7 cells infected with *M. smegmatis* were treated with methylprednisolone (10^2^ μM), BAY 11-7082 (2 μM), and BCI (1 μM) for 48 h, and the cell supernatant was collected for enzyme linked immunosorbent assay. (**A**) IL-6 secretion level, (**B**) TNF-α secretion level. The analysis of variance was used for data analysis, *** *p* < 0.001, **** *p* < 0.0001, ^ns^ no statistical differences.

**Table 1 microorganisms-11-00768-t001:** Oligonucleotides used in this study.

Primer Sets Name	Real-Time Quantitative PCR Primer (5′ to 3′)
GAPDH	F:AGGTCGGTGTGAACGGATTTG R:GGGGTCGTTGATGGCAACA
DUSP1	F:TGTTGTTGGATTGTCGCTCCT R:TTGGGCACGATATGCTCCAG

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
