# Peer review of "Methylprednisolone Promotes Mycobacterium smegmatis Survival in Macrophages through NF-κB/DUSP1 Pathway"

_microorganisms, 2023, doi:10.3390/microorganisms11030768_

Round 1

Reviewer 1 Report

Methylprednisolone suppresses cellular ROS production and IL-6 secretion to facilitate mycobacterial survival in macrophages through NF-κB/DUSP1 pathway

Dear author and editor:

The author in this work studied how Methylprednisolone suppress mycobacterial survival in macrophages, and suggested BCI as a new molecule for a host directed therapy in tuberculosis. the article could be published in microorganisms after a minor revision.

I have some comments on it:

·        It is well known that granuloma is the niche where the latent bacilli can survive. Why didn’t you try to study the effect of methylprednisolone on the granuloma like model ? and then see the efficacy of  BCI as host directed therapy.

Thank you very much, best regards

Author Response

Dear reviewer ,Please see the attachment. Thank you so much.

Reviewer 2 Report

In the article colleagues have introduced current data.

The following items should be corrected:

-              The title of the article is not clear.

-              The structure of the abstract should be corrected. Colleagues should write a structured summary including background; objectives; materials and methods; results; limitations and conclusions

-          It is recommended to structure the article, presenting the aim of the study, characteristic of patients in the materials and methods with a description of the design of the study.

-          How was diagnosis of LTBI established? This methodology must be clearly written.

-          Statistical methods have not presented.

-          Conclusions are not clear. Colleagues did not provide statistically significant evidence of conclusion data, moreover, the previous data was not defined.

All these concerns should be well addressed to consider this manuscript suitable for publication.

Round 2

Reviewer 2 Report

Dear colleagues,

The following items should be corrected:

The title, the objective and the conclusion of the article are not clear.

Author Response

(The authors gave the same response as above.)
